# Effective Removal of Boron from Aqueous Solutions by Inorganic Adsorbents: A Review

**DOI:** 10.3390/molecules29010059

**Published:** 2023-12-21

**Authors:** Xiang-Yang Lou, Lucia Yohai, Roberto Boada, Montserrat Resina-Gallego, Dong Han, Manuel Valiente

**Affiliations:** 1Grup de Tècniques de Separació en Química (GTS-UAB Research Group), Department of Chemistry, Facultat de Ciències, Universitat Autònoma de Barcelona, 08193 Bellaterra, Spain; xiangyang.lou@uab.cat (X.-Y.L.); yohai@fi.mdp.edu.ar (L.Y.); montserrat.resina@uab.cat (M.R.-G.); dong.han@uab.cat (D.H.); manuel.valiente@uab.cat (M.V.); 2Instituto de Investigaciones en Ciencia y Tecnología de Materiales (INTEMA), Universidad Nacional de Mar del Plata-Consejo Nacional de Investigaciones Científicas y Técnicas (UNMdP-CONICET), Mar del Plata B7608FDQ, Argentina

**Keywords:** boron, adsorption, inorganic adsorbents, surface modification, metal (hydr)oxides

## Abstract

Increasing levels of boron in water exceeding acceptable thresholds have triggered concerns regarding environmental pollution and adverse health effects. In response, significant efforts are being made to develop new adsorbents for the removal of boron from contaminated water. Among the various materials proposed, inorganic adsorbents have emerged as promising materials due to their chemical, thermal, and mechanical stability. This review aims to comprehensively examine recent advances made in the development of inorganic adsorbents for the efficient removal of boron from water. Firstly, the adsorption performance of the most used adsorbents, such as magnesium, iron, aluminum, and individual and mixed oxides, are summarized. Subsequently, diverse functionalization methods aimed at enhancing boron adsorption capacity and selectivity are carefully analyzed. Lastly, challenges and future perspectives in this field are highlighted to guide the development of innovative high-performance adsorbents and adsorption systems, ultimately leading to a reduction in boron pollution.

## 1. Introduction

Boron (B) is a non-metallic element that is widely found on the surface of the earth and in water in various forms, such as boric acid and borates [1]. Its average concentration in the environment varies across different media. In soil, the average concentration of boron is approximately 30 mg/kg; in seawater, it is approximately 4.5 mg/L; in groundwater, it can range from 0.3 to 100 mg/L [2]. This widespread presence of boron underscores its significance and ubiquity in the natural environment.

### 1.1. Boron Sources and Related Problems

Boron is introduced into the environment through a combination of natural phenomena and human activities [3,4]. Naturally occurring boron compounds originate from volcanic gases and hot springs near volcanic active areas. These compounds include borosilicate, boroaluminosilicate, and borate minerals [5]. In addition, boron is released into the environment through the weathering of rocks and leaching of salt deposits [6]. Furthermore, it is noteworthy that boron can be found naturally in seawater [7]. The concentration of boron in seawater varies significantly depending on the geography and location of ocean bodies. For instance, in the Mediterranean Sea, boron concentrations can reach as high as 9.6 mg/L [8]. Because of the geochemical nature of the drainage area, boron is also found in rainfall in coastal areas [9]. Industrial activity is another important source of boron released into the environment [10,11]. Boric acid and boron salts are widely utilized for manufacturing [12,13], with the glass industry alone consuming over half of the total boron compounds produced [14].

The boron released into the environment exhibits high volatility. It can evaporate, forming acid rain that falls back to the ground, contaminating drinking water, accumulating in the soil, and being taken up by plants, resulting in a series of environmental and health issues [15]. Indeed, the concentration range between boron deficiency and toxicity is extremely narrow, and this means that even a slight increase in boron levels can be hazardous to certain organisms. Unfortunately, the prevalence of toxic effects due to excess boron is more common than boron deficiency in the environment [16,17]. Hence, despite its essential role in plant growth, it is necessary to control the boron concentration in both drinking and irrigation water. The World Health Organization (WHO) established a guideline for boron, limiting it to 2.4 mg/L in drinking water and 1.0 mg/L for irrigation water to avoid damaging sensitive crops [18]. In addition, many countries have established regulations. For example, the boron threshold for drinking water in the European Union was recently set to 1.5 mg/L, whereas 1.0 mg/L was established in the United Kingdom, South Korea, and Japan, and the limit in the United States ranges between 0.6 and 1.0 mg/L depending on the state [19].

### 1.2. Chemistry of Boron in Aqueous Solution

In aqueous solutions, boron usually exists in the form of boric acid, B(OH)_3_, and various kinds of borates, depending on the pH and the concentration [20]. The distribution of borate species in solution under different pH values and at different initial concentrations was performed using Hydra-Medusa software v1.0 [21,22]. B(OH)_3_ and B(OH)_4_^−^ are mainly present at low boron concentrations (<216 mg/L), as shown in Figure 1. At high boron concentrations (>270 mg/L), and with an increase in pH value from 6–10, water-soluble polyborate ions such as B_2_(OH)_5_^−^, B_3_O_3_(OH)_4_^−^, and B_4_O_5_(OH)_4_^2−^ are formed [23]. The formation of these polynuclear ions is attributed to the interaction of boric acid and borate ions in solution. In general, regardless of the concentration, boric acid dominates at low pH, while borate ions dominate at high pH.

### 1.3. Boron Removal Methods

The development and investigation of efficient boron removal processes from aqueous solutions have become a crucial task due to the increasing boron concentration in surface and groundwater, as well as the need to treat seawater in desalination plants and wastewater from industries using boric acid as raw material [24,25,26]. The major challenge in selectively removing boron lies in the fact that it exists in water in the forms of various chemical species and that their concentrations vary from one region to another [27].

In past decades, a variety of methodologies such as precipitation [28], extraction [29], ion exchange [30], adsorption [31], and membrane [32] processes have been studied for boron removal [33]. The reported results indicate that boron is much more difficult to remove than many metalloids. For instance, the coagulation, flocculation, and filtration techniques that are frequently used for water purification offer little effectiveness in boron removal [24]. Reverse osmosis (RO) exhibits poor boron rejection primarily because boron, being a small species and undissociated acid, cannot be effectively rejected by the membrane owing to its lack of charge [34]. Many advances have been introduced to membrane processes to reach a removal rate higher than 90%; however, this also increased the operational time, cost, and system complexity [14]. Commercial chelating resins with functional groups have also been reported for effective boron removal through ion exchange with high selectivity [30,35]. The boron adsorption capacities of some commercial resins, modified resins, and other modified adsorbents are displayed in Table 1. However, their poor chemical and thermal stability, as well as their high price, easy fouling, and high recycling cost, severely limit their practical applications.

In contrast, adsorption offers several advantages over the aforementioned techniques for removing boron from water [40]. It is easy to implement, requires no additional chemicals, and is relatively inexpensive to operate. Some adsorbents can be regenerated multiple times with proper treatment, significantly reducing the cost per unit of treated water compared to other processes. Additionally, adsorption is an efficient process capable of removing substantial quantities of boron in a short time. Among the traditional adsorbents, activated carbon has been widely used in many applications; however, it has not been extensively used for boron removal because of its poorly performance for adsorbing polar adsorbates, such as boron, which confers a low effective surface active area, causing low boron adsorption selectivity unless modified [19,41,42]. In contrast, nanostructured and porous inorganic materials have emerged as the most promising materials due to their large specific surface areas and stable structures [43,44]. Recent efforts have been devoted to the development of novel low-cost inorganic adsorbents with high adsorption capacities. These research findings have significantly promoted the development of adsorption techniques for effective boron removal from water.

Herein, a comprehensive analysis of the recent advances in boron adsorption from aqueous solutions using inorganic adsorbents is provided. The review begins by presenting an overview of numerous adsorbents reported in recent years. Subsequently, a careful review of various functionalization strategies aimed at enhancing both boron adsorption capacity and selectivity is presented. Lastly, the current challenges and future perspectives in this field have been outlined to guide the development of next-generation, high-performance adsorbent materials and adsorption systems for boron removal.

## 2. Inorganic Adsorbents for Boron Removal

Among all the adsorbents used for boron adsorption, inorganic adsorbents like aluminum, iron, and magnesium oxides are of great interest because of their stability in a wide range of temperatures and times of exposure; in addition, they have low cost, are easy to synthesize, and could provide a moderate to large specific surface area depending on the porosity or surface to volume ratio of the nanostructured materials. Table 2 shows a summary of the boron adsorption performance of some inorganic adsorbents. It is worth mentioning that numerous synthesis methodologies for obtaining inorganic adsorbents such as sol-gel, gas phase, coprecipitation, hydrothermal synthesis, and microemulsion reaction, among others, have been reported [45]; however, they will not be extensively discussed here.

### 2.1. Aluminum Oxide

Aluminum oxide (Al_2_O_3_) is a versatile and widely used material due to its good mechanical properties, physicochemical stability, corrosion resistance, and high specific surface area (150–280 m^2^/g) when presented as a nanostructured porous material. Its great adsorption affinity towards inorganic compounds has made alumina one of the most studied materials in water treatment technologies for wastewater treatment or groundwater remediation [67].

The optimum pH for the removal of boron by alumina is found in the range of 7 to 9, normally with maximum adsorption observed at pH 8 for both amorphous and crystalline alumina [46,47,68]. The pH level aligns closely with the pH_pzc_ value of alumina (8.0–9.0), slightly lower than the pKa of boric acid (9.2). This indicates that the most favorable conditions for efficient boron removal occur when the adsorbent surfaces are uncharged and boric acid is the predominant boron species in the solution. In most of the reported studies, researchers worked with initial concentrations lower than 270 mg/L to avoid the appearance of polyborates such as B_2_O(OH)_6_^2−^, B_3_O_3_(OH)_4_^−^, B_4_O_5_(OH)_4_^2−^, and B_5_O_6_(OH)_4_^−^ [69,70]. Additionally, some studies have demonstrated that ionic strength has a negligible effect on the adsorption process, suggesting that electrostatic interactions are of minor importance for alumina and that adsorption occurs mainly via inner-sphere complex formation [48].

Boron chemistry and reported studies suggest that boron adsorption onto alumina prefers to be carried out at pH 8 and room temperature (RT). This pH is not only the one with maximum adsorption capacity but is also highly representative of surface and underground waters, which is an advantage when designing a purification or desalination process. In this way, Bouguerra et al. [47] used activated alumina, for which they found that for initial concentrations of 5 and 50 mg/L, the uptake of boron reaches approximately 40% and 65%, respectively, with an adsorbent dose of 0.8 and 5.0 g per 100 mL, and a contact time of 0.5 h. In terms of adsorption capacities, these values are 0.25 and 0.65 mg/g, respectively. In the case of Konstantinou et al. [46], where the adsorption mechanism is based on inner-sphere complexation, the maximum adsorption capacity for alumina was 0.4 mg/g. The maximum removal efficiency for alumina was obtained at a relatively low boron concentration and with a high contact time of 72 h. Similar results were obtained by Demetriou et al. [49], with a maximum adsorption capacity of 0.43 mg/g, represented via inner-sphere complex formation. However, in the study of Demey et al. [50], the adsorption capacity reached 4.41 mg/g at pH 8 after 72 h of contact time; this capacity is around 10 times that of previous works. One of the latest studies with nano-γ-Al_2_O_3_, prepared from aluminum foils, reached a boron adsorption capacity of 25.86 mg/g at 35 °C, 24 h of contact time, and pH 10 [51]. In addition, Lou et al. [48] reported a hierarchical alumina microspheres sorbent synthesized via a microwave-assisted coprecipitation method. The adsorption capacity was 51.60 mg/g at an initial boron concentration of 800 mg/L, while the theoretically calculated maximum adsorption capacity using the Langmuir model reached 138.50 mg/g. This outstanding performance can be primarily attributed to the microporosity of the hierarchical dandelion-like structure of the adsorbent.

### 2.2. Iron Oxide

Iron oxide materials have been intensively studied in recent decades because of their properties to be used in fields like catalysis, biomedical applications, gas sensing materials, electromagnetism, and water purification [71]. Some materials can adsorb contaminants in water, such as heavy metal ions, organic compounds, and other contaminants like arsenic and boron [72]. However, the applications of iron oxides strongly depend on the type of oxide and the stability of the particles under specific conditions. Sixteen pure phases of iron (hydr)oxides, i.e., oxides, hydroxides, and oxyhydroxides, are known to date [73]. Common synthetic methods for magnetic nanoparticles (MNPs) encompass coprecipitation, microemulsion, thermal decomposition, sol-gel, and solvothermal/hydrothermal processes [74]. Among all, coprecipitation is currently the most widely used method, and its main advantage is that a single synthesis can produce large quantities of products, and the reaction time is short, which is suitable for large-scale industrial production. However, it is still a challenge to regulate particle size and avoid agglomeration [72]. Another approach proposed by Glasgow et al. [75] is the synthesis of crystalline and uniform Fe_3_O_4_ through the thermal decomposition of iron precursors in a continuous flow method. Nevertheless, the Fe_2_O_3_/γ-Fe_2_O_3_ phase might be present together with Fe_3_O_4_.

The most common magnetic nanoparticles used in boron removal are magnetite (Fe_3_O_4_), maghemite (γ-Fe_2_O_3_), and some iron oxide-hydroxides such as Fe(O)OH. γ-Fe_2_O_3_ has been studied for boron adsorption, demonstrating peak adsorption capacity at pH levels ranging from 8 to 9. On the other hand, Fe_3_O_4_ possesses strong magnetic characteristics, enabling straightforward separation from the reaction medium after adsorption via a simple magnet [72]. Demetriou et al. [52] studied boron adsorption using iron(III) oxide hydroxide (Fe(O)OH). The maximum adsorption capacity was found to be 0.32 mg/g at pH 8 after 72 h of contact time. Thermodynamic values and a lack of ionic strength dependence suggest that adsorption is based on inner-sphere complexation, as found for alumina adsorbents. Liu et al. [53] studied Fe_3_O_4_ and two different surface modification adsorbents derived from it. They found that, for all MNPs, boron is adsorbed in the forms of both H_3_BO_3_ and B(OH)_4_^−^. Furthermore, it was observed that in all cases, the highest level of boron adsorption occurred in neutral solutions, likely attributed to the presence of hydrogen bonding, electrostatic attractions, and hydrophobic interactions. Conversely, the lowest adsorption was observed in basic solutions due to electrostatic repulsion. Additionally, the study demonstrated a decrease in adsorption as ionic strength increased. In another study, Chen et al. [54] reached equilibrium rapidly in 1.5 h, showing a high capacity of 49.41 mg/g at pH 7 and 45 °C. Surface analysis indicated the formation of new Fe-O-B bonds in this case.

Despite their versatility and wide range of applications, iron oxides are frequently used after surface modification and/or functionalization to increase their affinity and selectivity towards boron [76]. Moreover, the surface modification of iron oxide magnetic nanoparticles serves to decrease surface energy, reduce agglomeration in solution, and enhance hydrophobic characteristics. The various surface modifications applied to boron adsorbents will be discussed in detail in the following section.

### 2.3. Magnesium Oxide

Magnesium oxide (MgO) has various applications in industry. The physical and chemical properties of magnesium oxide particles, including their morphology, are primarily governed by the precursor used for its synthesis. It has photocatalytic properties, and it has been tested as an adsorbent for toxic chemical agents [77]. Indeed, magnesium oxide is an effective and low-cost adsorbent for the removal of boron from aqueous solutions. Chemisorption on MgO can be primarily ascribed to the acid/base interaction of adsorbates with its surface. The reported pH_PZC_ for MgO lies between 9.8 and 12.7; consequently, most studies have been carried out at pH under 10 to have a positively charged surface. For example, Kameda et al. [55] attributed boron adsorption on the MgO surface to the electrostatic attractive force of positively charged MgO with B(OH)_4_^−^. The value of the maximum adsorption was 232.4 mg/g at 30 °C and a week of contact time following a pseudo-first-order reaction kinetics and a Langmuir-type adsorption isotherm. Boron desorption from MgO particles containing adsorbed B(OH)_4_^−^ species was possible using a 0.1 M NaOH solution as a rinse; however, the adsorption capacity of regenerated MgO was lower than that of the original MgO.

Li et al. [56] used MgO nanosheets for boron adsorption. The MgO nanosheets were fabricated by an ultrasonic method by magnesium salt solution precipitation, calcination, and further ultrasonic treatment. Adsorption capacity increased with the temperature reaching the maximum, 87 mg/g, at 90 °C. The results were better described by the Langmuir model than the Freundlich model. Characterization techniques have shown that both the hydration of MgO nanosheets and the polymerization of adsorbed boron species occur concurrently, leading to the coexistence of unbound surface H_2_O molecules.

Masindi et al. [57] investigated boron adsorption using calcinated magnesite tailings. In contrast to other reported results, the removal yield remained nearly constant across the entire pH range of 1.0–8.0, not reaching a maximum at the highest pH. At pH values higher than 8, boron removal was reduced. The adsorption process took 30 min to reach equilibrium and obtained a capacity of 6 mg/g at pH 8. Kinetic and isotherm data were best modeled by the pseudo-second-order and Langmuir isotherm equations, respectively.

The research of De la Fuente et al. [58] showed that the removal rate improved as the pH increased, presenting a maximum adsorption capacity at a pH value between 9.5 and 10.5. For an initial concentration of 50 mg/L, the maximum adsorption capacity was obtained after 30 min of stirring and 6 h of repose at 70 °C. Under these optimum conditions, a boron removal rate of around 95% was obtained; however, this value decreased to 75% at 30 °C. In a subsequent article [78], they demonstrated that the adsorption process followed a pseudo-second-order kinetic model and the Langmuir isotherm model. Nevertheless, the reaction process between boron and magnesium compounds was found to be irreversible, and neither pH nor temperature adjustment could be used to regenerate the adsorbents. In their study, Fukuda et al. [59] developed a low crystalline magnesium oxide material (LC-MgO), exhibiting a remarkable boron adsorption capability of 27.3 mg/g with a 60 min contact time at pH 7 and with an initial boron concentration of 500 mg/L. This behavior was attributed to the formation of Mg-B layered compounds that precipitated on the MgO particle surface. One of the last works regarding boron adsorption onto MgO corresponds to Song et al. [60], where MgO was previously hydrated to form Mg(OH)_2_, responsible for boron adsorption. This material exhibited the highest adsorption capacity, reaching 202.4 mg/g in 6 h at 25 °C and pH 9, and exhibited a 70% surface regeneration after treating it with NaOH.

### 2.4. Mixed Oxides

The use of metal oxide mixtures has attracted wider attention due to the possibility of achieving synergistic effects of the mixed metal oxides to develop better properties and achieve improved performance than individual oxides.

Mixed oxides have also been widely reported for boron adsorption [5]. As Guan et al. [19] highlighted, the boron adsorption capacity of layered double hydroxides (LDHs) can be enhanced through thermal activation via calcination at high temperatures. This process alters the LDHs’ structure by removing interlayer anions and water, promoting the formation of a mixed metal oxide material. Various LDHs have been utilized for effective boron removal. The most important works in terms of adsorption capacity are the ones from Gao et al. [61], Liu et al. [62], Kentjono et al. [63], Kurashina et al. [64], and Kameda et al. [65].

In the study by Gao et al. [61] on ultrathin Mg-Al LDH, the maximum adsorption capacity was 77.8 mg/g at 25 °C and pH 7. However, it is worth noting that the initial boron concentration used was considerably lower than in other studies (see Table 2). Liu et al. [62] synthesized Mg-Al LDH with the highest adsorption capacity of 33 mg/g. Kentjono et al. [63] found that Mg-Al (NO_3_) LDH exhibited a notable boron removal efficiency (37.9 mg/g) when optoelectronic wastewater was treated at pH 9.0–9.2. Kurashina et al. [64] employed Mg-Al LDH to treat a highly concentrated borate solution, achieving a 90% elimination rate at pH 10 with an adsorption capacity of 25 mg/g. Kameda et al. [65] conducted a comparative study on two Mg-Al LDHs intercalated Cl^−^ and NO_3_^−^, respectively, for boron uptake. They found that the maximum adsorption values were 38.9 mg/g for NO_3_^−^·Mg-Al LDH and 41.1 mg/g for Cl^−^·Mg-Al LDH, respectively. All these works reached a similar adsorption capacity which can be due to the presence of hydroxyl groups on the surface that can provide sites for boron adsorption through surface complexation [79].

In other works, tertiary mixed oxides were studied. Heredia et al. [78] studied boron removal capacity from aqueous solutions utilizing MgAlFe mixed oxides. They achieved a maximum boron removal exceeding 85%, equivalent to 9.08 mg/g. They attributed boron adsorption to electrostatic attraction between the surface of mixed oxides and boron species. Other tertiary oxides focused on aluminum-based oxides were approached by Irawan et al. [66]. They studied three samples of aluminum-based residuals (Al-WTR1, Al-WTR2, and Al-WTR3) composed of Al_2_O_3_, Fe_2_O_3_, and SiO_2_ in different ratios. The optimum pH was found at 8.2–8.5, and maximum adsorption capacities were 0.980, 0.700, and 0.190 mg/g, respectively, relatively low compared to Mg-Al LDH. The adsorption reaction was spontaneous and exothermic, and it was characterized by physical adsorption (electrostatic interactions and van der Waals forces).

Despite the extensive reporting of synthetic and naturally occurring mixed oxides as effective boron adsorbents, practical applications still face limitations. In the case of synthetic mixed oxides, a significant drawback is the inability to exercise precise control over the synthesis process that guarantees the same resulting product. This constraint arises from direct dependence on the proportions of constituent elements and the initial reaction conditions. Furthermore, the understanding of the surface chemistry of mixed oxides may be insufficient, leading to challenges in predicting and optimizing their adsorption behavior, especially with the existence of competing anions such as CO_3_^2−^, Cl^−^, and NO_3_^−^ [79]. Natural mixed oxides, on the other hand, face the primary challenge of heterogeneity and varying availability across different geographical locations. Consequently, despite their advantageous properties, obtaining uniform quantities of natural mixed oxides can prove to be a challenging task.

## 3. Modification of the Inorganic Boron Adsorbent

In recent studies, inorganic adsorbents have been extensively utilized due to their distinct advantages and outstanding performance. However, these adsorbents face a significant drawback in their relatively lower boron adsorption capacities compared to other metallic ions. The negative charge of boron species in natural waters requires the design of adsorbents with positive surfaces, enabling physical electrostatic interactions to enhance boron adsorption. More importantly, the primary driving force in chemical boron adsorption is the formation of surface complexations through ion/ligand exchange between adsorbents and boron ions [48]. Consequently, research efforts have been focused on exploring various surface modification strategies using different types of molecules to maximize boron adsorption.

### 3.1. Functional Groups Used for Boron Adsorbent Modification

According to the literature, molecules containing three or more hydroxyl groups located in the cis position, known as “vis-diols”, tend to form stable complexes with boron species in both acidic and alkaline environments [19,80]. This suggests a promising functionalization strategy for the specific removal of boron species from water, involving the utilization of polyhydroxy groups as chelating agents. Previous studies have successfully employed sugar derivatives for boron removal [81,82]. These polyoxide compounds typically bind by forming boric acid esters or borate anion complexes with a proton as a counterion. Consistent with these results, tertiary amine groups are crucial for chelating boron because they can bind a proton that is released when borate is complexed with hydroxyl functionalities [83,84,85]. *N*-methyl-d-glucamine (NMDG), glucose, iminodipropylene glycol, polyalcohols, polyether sulfonates, hydrogels, and other similar substances are used to functionalize a variety of modified adsorbents based on this theory, as summarized in Table 3.

Among all types of molecules, NMDG has been mostly studied. The presence of polyols and tertiary amine ends in NMDG can increase the possibility of complexation reactions with boron. The binding process of boron with NMDG-modified adsorbents is shown in Figure 2. Using the click coupling technique, Tural et al. [86] synthesized magnetic nanoparticles modified with NMDG. The maximum adsorption capacity was 13.44 mg/g, which was significantly higher than the adsorbent synthesized by direct coupling. The boron adsorption kinetics followed a pseudo-second-order model, and the adsorption was consistent with the Langmuir isotherm. Hong et al. [97]. reported an NMDG-modified cellulose acetate as an effective adsorbent for boron removal from aqueous solutions. The adsorbent capacity reached ~34 mg/g within 10 min, which overcame the disadvantage of the slow adsorption rate observed with the current commercial resin (Amberlite IRA743).

In addition to the extensive use of NMDG modifications, numerous other useful molecules have also been employed. Oladipo et al. [87] reported glycidol-modified magnetic chitosan beads (MCG) with cis-diol groups. After modification, the maximum adsorption capacity increased from 66.85 mg/g to 128.50 mg/g. MCG micro-spheres were easily separated from the solution and could be reused up to seven times. Even in the presence of competing ions such as Cu^2+^, Fe^3+^, Ni^2+^, Mg^2+^, Ca^2+^, Na^+^, and K^+^ ions, MCG still showed approximately 96% removal efficiency from the boric acid solution (125 mg/L). Their other work [88] reported functionalized magnesium ferrite magnetic nanopowder with polyvinyl alcohol and glycidol as boron-selective adsorbents. For polyvinyl alcohol-functionalized particles, the adsorption peaked at pH 7.0, while for glycidol-functionalized particles, the adsorption peaked at pH 10.0. At a low boron concentration of 5 mg/L, polyvinyl alcohol-functionalized polymers showed a higher adsorption capacity (45.5 mg/g). However, at a higher boron concentration (50 mg/L), glycidol-functionalized particles showed a higher adsorption capacity of 68.9 mg/g. Demey et al. [50] synthesized a new composite adsorbent consisting of alginate-alumina (CAAl) for boron removal. After modification, the alumina showed an increased maximum experimental adsorption capacity from 0.41 to 1.81 mg/g at pH 9.5. The experimental results of the equilibrium data were better fitted by the Langmuir equation, and the kinetic data followed a pseudo-second-order rate equation. The incorporation of alumina particles in the new alginate-based composite improves mechanical strength and slightly improves boron removal. Sanfeliu et al. [89] studied the removal of boron using nanosized mesoporous silica that had been crosslinked in successive steps with 3-aminopropyltriethoxysilane (APTES) and polyhydroxy compounds (gluconolactone), respectively. The maximum adsorption capacity obtained was 20.0 mg/g. Each anchored glucose group had multiple boron coordination, according to the results obtained. Under ambient conditions at pH 7, Tang et al. [90] investigated a group of diol-functionalized silica particles for boron removal. The adsorption capacity was less than 5 mg/g. The adsorption data were well fitted by the Sip’s isotherm model and the pseudo-second-order kinetic model.

### 3.2. Methodologies Used for Boron Adsorbent Modification

The chemical properties of different functional groups require the use of diverse adsorbent modification techniques. This necessity arises from the varying levels of hydrophobicity and polarity exhibited by these functional groups, which, in turn, impact their interactions with adsorbent materials. The most frequently reported methods are suspension polymerization reactions, as well as polymerization or copolymerization reactions [98,99].

For instance, Adeyemi et al. [87] studied the modification of magnetic chitosan beads using glycidol. The process involved dispersing magnetic chitosan beads in ethanol, followed by the addition of a glycidol solution in *N*-methyl-2-pyrrolidone to the reaction flask. The suspension was then stirred for 10 h, and the final product was collected, washed with ethanol, and dried under vacuum at 70 °C. Similar modification approaches were reported by Liao et al. [100] and Wu et al. [101].

However, direct bonding to inorganic adsorbents such as alumina, iron oxide, and magnesium oxide may not always be practical, especially when using NMDG as the functional group. In such situations, organosiloxane groups can serve as a bridge between NMDG and inorganic adsorbents. Two primary strategies have been reported to achieve this objective.

One strategy involves modifying the surface of inorganic adsorbents with halogen or epoxy groups through silanization reactions and subsequently linking the NMDG functional groups via nucleophilic substitution reactions. However, this synthetic route imposes strict requirements on the pH of the solution because the functional groups contain a high number of amino groups. Excessive alkalinity (pH > 11) can result in the dissolution of the silica gel alkyl. The use of a single low-polarity solvent (e.g., dioxane) or a mixed solvent (e.g., dioxane and water) can reduce the solution’s alkalinity (pH < 10) but may also hinder the effective progress of nucleophilic substitution reactions of halogen, epoxy groups, and functional groups. Tural et al. [91] used a two-step synthesis to functionalize γ-Fe_2_O_3_-silica particles with NMDG. In the first step, γ-Fe_2_O_3_-silica composites were functionalized with trimethoxysilylpropyl chloride. Subsequently, the obtained magnetic composite reacted with NMDG to introduce tertiary amine functionality into the material. A schematic illustration of these two steps for functionalizing γ-Fe_2_O_3_-silica particles with NMDG is shown in Figure 3.

Another strategy for connecting NMDG to inorganic adsorbents is to first form a modified silane coupling agent via the reaction of functional groups with halogen and epoxy groups, as shown in Figure 4. In this method, a silica gel coupling agent is generated to reduce the alkalinity of the reaction system and avoid the dissolution of silica gel, thereby improving the synthesis efficiency. However, this method results in a wide particle size distribution of the synthesized adsorbent. Xu et al. [92] successfully utilized this strategy to synthesize NMDG-modified silica-supported adsorbents. They synthesized NMDG-modified GPTMS (Si-NMDG) and then loaded it onto an activated silica surface via salinization.

## 4. Challenges and Perspectives

Achieving efficient boron adsorption from water is significantly important for environmental protection, safeguarding human health, and resource recovery. Despite the encouraging advances made in recent years in the development of new adsorbents, there are still several issues that must be addressed and overcome before these adsorbents become a viable option for commercial use. In this section, the challenges and future perspectives of boron adsorption in water treatment systems are outlined.

Firstly, current adsorbents are limited by their low selectivity and capacity, requiring large quantities of adsorbents to achieve the desired removal efficiency. Moreover, their high cost, powdered form, and challenging recovery hamper their practical applications. Therefore, it is crucial to focus on the development of new adsorbents with high adsorption capacity, selectivity, recyclability, affordability, and scalable production in the future. Secondly, although ion/ligand exchange interactions between adsorbents and boron species are frequently reported, there are limited investigations of the underlying adsorption mechanism at the molecular level. The use of characterization techniques that provide a molecular picture of the absorbate-surface interaction (e.g., molecular spectroscopic techniques, density functional theory (DFT), and molecular dynamics simulations) is encouraged to improve the understanding of the adsorption and desorption processes. Finally, it is suggested that potential synergies between adsorption and other technologies, such as membrane processes, be explored to address the complexities of certain water systems and improve outcomes. Thus, there is a wide scope for future research to broaden our knowledge about the boron adsorption and desorption mechanisms and to promote their applications in water treatment.

## Figures and Tables

**Figure 1 molecules-29-00059-f001:**
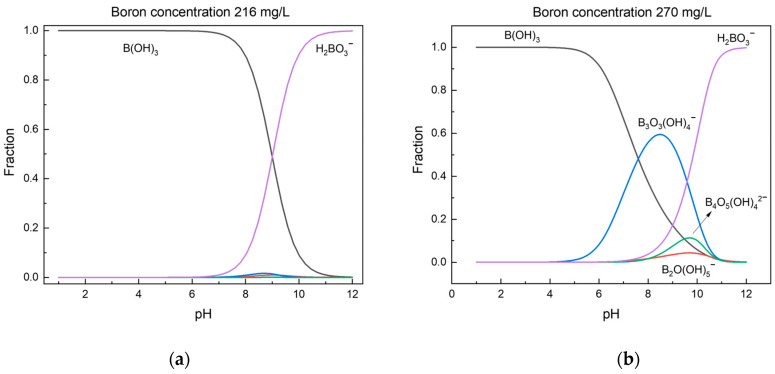
Distribution of borate species in solution as a function of pH, depicted for concentrations of 216 mg/L (**a**) and 270 mg/L (**b**).

**Figure 2 molecules-29-00059-f002:**
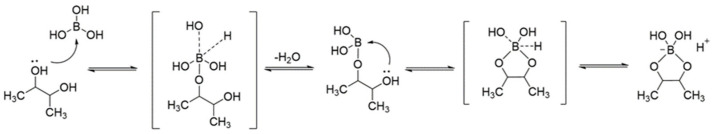
Example of the binding mechanism of boron with any glycol compound used as adsorbent. Adapted with permission from Ref. [83]. Copyright 2022, copyright Elsevier Ltd.

**Figure 3 molecules-29-00059-f003:**
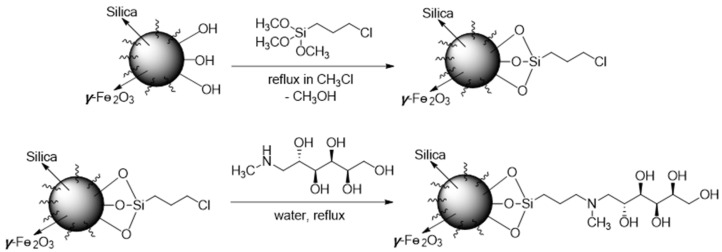
A schematic illustration of preparation steps for functionalizing γ-Fe_2_O_3_-silica particles with NMDG. Adapted with permission from Ref. [91]. Copyright 2010 copyright John Wiley and Sons.

**Figure 4 molecules-29-00059-f004:**
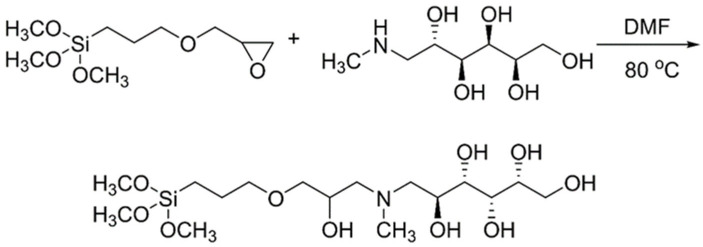
Synthesis pathway of organosiloxane Si-NMDG. Adapted with permission from Ref. [93]. Copyright 2016, copyright Wiley-VCH Verlag GmbH & Co. KGaA.

**Table 1 molecules-29-00059-t001:** Boron adsorption capacities and particle size of some commercial resins, modified resins, and other modified commercial adsorbents.

Adsorbents	Adsorption Capacity (mg/g)	Particle Size (μm)	Reference
CL-RESIN	8.4	722–855	[36]
NCL-RESIN	8.6	710–845	[36]
IRA 743-RESIN	10.9	550–700	[36]
P(GMA-co-TRIM)-EN-PG	29.2	<241	[37]
P(GMA-co-TRIM)-TETA-PG	23.3	<273	[37]
CL-MCM-41	19.5	-	[38]
NCL-MCM-41	16.7	-	[38]
T-RESIN	21.3	-	[39]

**Table 2 molecules-29-00059-t002:** Summary of boron adsorption capacities of different inorganic adsorbents.

Adsorbent	Surface Area (m^2^/g)	Adsorption Capacity (mg/g)	Temperature (°C)	Contact Time (h)	pH	Ref.
Al_2_O_3_	150	0.35 at 15 mg/L B concentration	25	72	8	[46]
	-	0.65 at 50 mg/L B concentration	25	0.5	8–8.5	[47]
	250	138.50 *	25	2	8.0	[48]
	169	0.43 *	22 ± 3		8.5	[49]
	-	6.38 *	20	72	8	[50]
	150	25.86 *	35	24	10	[51]
Fe(O)OH	-	0.32 *	22	72	8	[52]
Fe_3_O_4_	-	0.26 at 21,622 mg/L B concentration	22	48	6	[53]
	107	49.41 at 7567.7 mg/L B concentration	45	1.5	7	[54]
MgO	4.8	232.44 *	60	24	<8	[55]
	168	87.03 *	90	3	10	[56]
	15	5.4 *	RT	0.5	<8	[57]
	-	54.2 *	RT	48	9.5–10.5	[58]
	165	27.3 at 500 mg/L B concentration	-	1	7	[59]
	280	202.43 *	25	6	9	[60]
Mixed oxides	213	77.84 *	25	-	7	[61]
	-	33 at 21,622 mg/L B concentration	90	3	9	[62]
	10	37.90 *	RT	4	9	[63]
	-	25 *	-	-	10	[64]
	-	41.08 *	30	2	10	[65]
	40.5	0.98 *	RT	24	8.2–8.5	[66]
	34.6	0.70 *				
	14.5	0.19 *				

* Maximum adsorption capacity obtained from fitting the adsorption isotherm.

**Table 3 molecules-29-00059-t003:** Boron adsorption capacity reported for some modified adsorbents.

Adsorbents	Adsorption Capacity (mg/g)	Isotherm and Adsorption Kinetic Models	Ref.
Magnetic nanoparticles attached to NMDG	13.4 *	Langmuir, pseudo-second-order	[86]
Glycidol-modified magnetic chitosan beads (MCG)	128.5 *	Redlich–Peterson, pseudo-second-order	[87]
Magnesium ferrite magnetic with polyvinyl alcohol	45.5 at 100 mg/L	-	[88]
Magnesium ferrite magnetic with glycidol	68.9 at 100 mg/L	-	[88]
Alginate–alumina (CAAl)	56.2 *	Langmuir, pseudo-second-order	[50]
Silica matrix with gluconamide moieties	20.0 *	Langmuir–Freundlich model	[89]
Diol-functionalized silica particles	57.09 *	Sips, pseudo-second-order	[90]
NMDG-modified magnetic microparticles	0.02 at 0.9 mg/L	-	[91]
Silica-supported NMDG	16.7 *	Freundlich, pseudo-second-order	[92]
Poly(Si-NMDG)@MIL-101(Cr)	24.8 *	Langmuir	[93]
Silica–polyallylamine composites grafting NMDG	16.8 *	Freundlich, chemical reaction	[94]
*N*-methyl-d-glucamine-based hybrid	21.8 *	Langmuir	[95]
Fe_3_O_4_@SiO_2_ functionalized with glycidol	25.6 *	Langmuir	[96]

* Maximum adsorption capacity obtained from fitting the adsorption isotherm.

## Data Availability

No new data were created or analyzed in this study. Data sharing is not applicable to this article.

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
