# Peer review of "Effective Removal of Boron from Aqueous Solutions by Inorganic Adsorbents: A Review"

_molecules, 2023, doi:10.3390/molecules29010059_

Round 1

Reviewer 1 Report

Comments and Suggestions for Authors

Machanism is required to be discussed. If machanism discussion about  boron removal is lack, similar research should be refer. This should encourage researchers focus on the discussion about  boron removal.

Author Response

Included in the PDF

Reviewer 2 Report

Comments and Suggestions for Authors

The submitted review is clearly focused and well written. It seems that some pieces of introduction and description of the boron chemical properties are of textbook-level and could be omitted, but overall the content is nice. Below are several points which, if considered by the authors, can improve the review quality.

1. In lines 19-20: I believe it is better to change the list to "magnesium, iron, and aluminum individual and mixed oxides" to avoid confusion with elementary metals.

2. In lines 65-66 and later in the text - please indicate whether the concentrations in mg/L correspond to boron or borate ions. (personally, I would prefer concentrations in mmol/L throughout the text to eliminate the ambiguity, but at least the meaning should be explained).

3. If figure 1 was adopted from a published report, the source should be indicated in the caption.

4. Please double-check the data in Table 1. First, it is quite strange to see the references to papers rather than manufacturers' data regarding the commercial materials. Second, the particle size given for CL-MCM-41-RESIN and NCL-MCM-41-RESIN (1.3 and 1.4 nm) corresponds actually to the pore width according to ref. 37. Moreover, the modification to prepare these materials is described in ref. 37, and I doubt that these materials can be called commercial. Finally, I am not sure that all of the mentioned materials can be called resins (at least, this is not used in ref. 37). 

5. What about the use of conventional adsorbents to remove boron compounds? Active carbons, silica, etc - in view of the review title, these should be at least mentioned, even if the major focus on Mg, Fe, and Al species was chosen by the authors.

6. In figure 2, I wonder why the caption says "NMGD type" - the structure corresponds to any glycol.

7. In line 395, modification with propyl chloride is mentioned, which is misleading (should be trimethoxysilylpropyl chloride?).

8. I believe, the Conclusions section can be safely removed, since the authors' opinion is well conducted in the Challenges and perspectives, and the Conclusions section as it is now just duplicates the abstract.

9. My final remark is very general, and I have to begin with a broad introduction.

My concern originates from the fact that the topics related to adsorption have become very popular and (seemingly) easy to implement in the laboratory. I have reviewed several papers on adsorption recently, and they all (with just a single exception!) were extremely poorly written. They followed a general scheme like: perform adsorption of something on something, describe the obtained data with several adsorption equation, select this with the highest R2, prepare a set of conventional plots and tables - and this is all. Many of those papers contained rough mistakes like plugging initial adsorbate concentrations rather than equilibrium ones in the adsorption isotherm equations. Unfortunately, at least one of those papers was accepted by the editor despite my numerous comments on such issues - and overall I have to say that many of the recent adsorption-related studies contain many mistakes.

I am writing this here because in my view a review can point at these issues, to at least catch the attention to them. Of course, the review authors cannot correct the mistakes in the original reports, but they can express their opinion.

For example, Table 2 in the review under consideration translates one of the common aberrations related to adsorption - the strongly different adsorption capacity reported for the same material under different conditions. However, several strongly different values are mixed here:

a) adsorption capacity - the highest amount of adsorbate which can be bound to the adsorbent. It commonly corresponds to saturated solution of the adsorbate (which is normally not the case in the experiments) - refer to the IUPAC definition at https://goldbook.iupac.org/terms/view/A00156;

b) limiting adsorption - an estimate of the adsorption capacity from an adsorption isotherm, in lieu of the data for the saturated adsorbate solution;

c) specific adsorption reached under very specific experimental conditions (initial adsorbate concentration, volume of its solution, amount of the adsorbent, contact duration, stirring, etc).

The adsorbent efficiency can be expressed (and compared) in terms of adsorption capacity or at least limiting adsorption, whereas comparison of the experimental specific adsorption is useless (although this parameter is important for practical applications under certain specified conditions).

I therefore advise the authors to critically reconsider the data in Tables 2 and 3 of the considered review and add some appropriate comments to the text, to underscore the meaning of the presented data.

Author Response

Included in the PDF

Reviewer 3 Report

Comments and Suggestions for Authors

The work needs correction. The sol-gel methods in boron removal should be desribed.

Comments on the Quality of English Language

I am not qualified to evaluate English Language.

Author Response

Included in the PDF

Round 2

Reviewer 3 Report

Comments and Suggestions for Authors

The paper can be published in this form.